# Cross-Reactive Immunity among Five Medically Important Mosquito-Borne Flaviviruses Related to Human Diseases

**DOI:** 10.3390/v14061213

**Published:** 2022-06-02

**Authors:** Baohua Hou, Hui Chen, Na Gao, Jing An

**Affiliations:** 1Department of Microbiology, School of Basic Medical Sciences, Capital Medical University, Beijing 100069, China; hbh1517@163.com (B.H.); anjing@ccmu.edu.cn (J.A.); 2Experimental Center for Basic Medical Teaching, School of Basic Medical Sciences, Capital Medical University, Beijing 100069, China; 3Center of Epilepsy, Beijing Institute for Brain Disorders, Beijing 100093, China

**Keywords:** cross-reactive immunity, Japanese encephalitis virus, dengue virus, Zika virus, West Nile virus, yellow fever virus

## Abstract

Flaviviruses cause a spectrum of potentially severe diseases. Most flaviviruses are transmitted by mosquitoes or ticks and are widely distributed all over the world. Among them, several mosquito-borne flaviviruses are co-epidemic, and the similarity of their antigenicity creates abundant cross-reactive immune responses which complicate their prevention and control. At present, only effective vaccines against yellow fever and Japanese encephalitis have been used clinically, while the optimal vaccines against other flavivirus diseases are still under development. The antibody-dependent enhancement generated by cross-reactive immune responses against different serotypes of dengue virus makes the development of the dengue fever vaccine a bottleneck. It has been proposed that the cross-reactive immunity elicited by prior infection of mosquito-borne flavivirus could also affect the outcome of the subsequent infection of heterologous flavivirus. In this review, we focused on five medically important flaviviruses, and rearranged and recapitulated their cross-reactive immunity in detail from the perspectives of serological experiments in vitro, animal experiments in vivo, and human cohort studies. We look forward to providing references and new insights for the research of flavivirus vaccines and specific prevention.

## 1. Introduction

Flaviviruses are mainly transmitted between humans and animals through blood-sucking arthropods including mosquitoes and ticks. At present, mosquito-borne flaviviruses are extremely widespread, of which dengue virus (DENV), yellow fever virus (YFV), Japanese encephalitis virus (JEV), West Nile virus (WNV), and Zika virus (ZIKV) are responsible for significant human morbidity and mortality. Particularly, ZIKV has been transmitted rapidly throughout the Americas in recent years and is one of the causes of congenital brain abnormalities [1]. Following the importation of WNV into the United States in 1999, the virus spread widely, posing a serious threat to global public health (according to the World Health Organization (WHO)).

Flaviviruses cause a spectrum of potentially severe diseases. Clinical studies [2] demonstrate that dengue fever patients with heterologous DENV infection history tend to develop severe dengue including dengue hemorrhagic fever (DHF) and dengue shock syndrome (DSS). This may be related to the phenomenon of antibody-dependent enhancement (ADE): with the sub-neutralizing but high-binding ability to heterologous DENV, the antibodies induced by the initial DENV infection can promote the infection of heterologous DENV through Fcγ receptor-mediated viral entry [3].

Akin to DENV, common mosquito-borne flaviviruses also elicit abundant cross-reactive immune responses and affect each other [4,5,6]. Mosquito-borne flaviviruses are mainly transmitted by *Aedes* mosquitoes (such as YFV, DENV, and ZIKV) or *Culex* mosquitoes (such as JEV and WNV). The insect vectors tremendously affect viral distribution, so epidemic regions of different flaviviruses overlap widely [7,8] (Figure 1). As the most rampant of the flaviviruses, DENV has been estimated to cause approximately 400 million infections per year, stretching across more than 100 countries in tropical and subtropical regions [9]. Other flaviviruses, such as ZIKV and YFV, are also endemic in the Americas and Africa, formulating the opportunity for a person to be infected by multiple flaviviruses [7]. Additionally, other vector-borne flaviviruses including tick-borne encephalitis virus and Usutu virus, which quickly spread across Europe in recent years, have been reported to present cross-reactive immunity with WNV [10,11]. The highly competent vectors, coupled with extensive overlaps among epidemic regions, greatly increase the difficulty in the prevention and control of flavivirus infections [12].

At present, the vaccines licensed for the prevention of flavivirus infections include yellow fever (YF) live attenuated vaccine (LAV) (YF-17D), Japanese encephalitis (JE) LAV (SA 14-14-2), JE inactivated vaccine (INV) (JE-VAX, JEBIK, IC51, etc.) [13], and tetravalent chimeric dengue LAV (dengvaxia). Vaccines against other flaviviruses are still under development. Therefore, figuring out their cross-reactive immunity is meaningful for the prevention and control of flavivirus infections that lack vaccines. This review will generally discuss the progress in research of cross-protection and cross-enhancement among various mosquito-borne flaviviruses.

## 2. The Antigenic Relationships among the Five Mosquito-Borne Flaviviruses

Viruses in the genus *Flavivirus* of the family *Flaviviridae* are enveloped viruses with a positive-sense single-stranded RNA genome, encoding 10 proteins—capsid (C), premembrane (prM), envelop (E), and nonstructural proteins 1–5 (NS1, NS2A, NS2B, NS3, NS4A, NS4B, NS5). The flaviviruses are antigenically related by sharing similar antigenic determinants on their C and E proteins [14,15]. According to the comparison conducted by Chang, et al., approximately 80% of the identity shared between ZIKV and other flaviviruses exists in NS3 and NS5 proteins [16].

We used available published proteomic data to align the sequences of both complete polyprotein and E protein among the five flaviviruses mentioned above (Table 1). The alignments revealed that the closest relatives are JEV and WNV, displaying 77% identity on polyprotein and 79% identity on E protein. The ZIKV strain is at an intermediate position among JEV, WNV, and DENV strains (54–58% identity). The sequence identity between JEV or WNV and ZIKV is greater than that between JEV or WNV and DENV. The YFV strain is the farthest from the other flaviviruses, with sequence identity ranging from 40–46%.

The structural similarity of flaviviruses provides a molecular basis for cross-reactive immunity. Structural approaches such as cryoelectron microscopy and X-ray crystallography figure out the epitopes of monoclonal antibodies toward flaviviruses [17]. Epitopes in the fusion loop (FL) region of envelop protein domain II (EDII) contribute to numerous cross-reactive antibodies toward different flaviviruses with variable potency and sensitivity [18], because of their highly conserved sequences. Antibodies toward the FL region are always weakly neutralizing and pose a risk of ADE because the hydrophobic FL residues are buried in the adjacent E monomer, and less accessible on the mature virions. In contrast, antibodies towards the EDIII lateral ridge are usually of more potency to strongly neutralize the viruses [19].

## 3. Cross-Reactive Immunity between DENV and ZIKV

### 3.1. The Impact of Prior DENV Immunity on ZIKV Infection

(1)In vitro serological experiment

DENV infection induced few cross-reactive neutralizing antibodies (NAbs) against ZIKV. Swanstrom, et al. revealed that 73% of dengue fever patients’ sera failed to cross-neutralize ZIKV, 18% cross-neutralized ZIKV slightly, and only 9% strongly [20]. Among the 16 dengue convalescent sera collected by Dejnirattisai, et al., only three cases showed appreciable neutralizing ability towards ZIKV [21].

Instead, sera or monoclonal antibodies of FL epitope derived from dengue fever convalescent patients could induce ADE of ZIKV in U937 cells [21]. In cynomolgus monkeys, Breitbach, et al. demonstrated that the DENV2–4 immune sera could not neutralize ZIKV, either, and only DENV1-immune sera presented ZIKV reactive Nabs [4].

(2)The in vivo mouse model

The DENV-immune background could cross-protect mice against ZIKV infection in a way other than cross-reactive antibodies. In the *ifnar^−/−^* Sv/129 mouse model used by Watanabe, et al., all DENV2 inoculated mice survived when challenged with ZIKV, while the passive transfer of DENV2-immune human or mouse sera to *ifnar^−/−^ifngr^−/−^* mice did not affect mortality and viremia levels when challenged with ZIKV [5]. Instead, Bardina, et al. detected higher viremia levels of ZIKV in *Stat2^−/−^* mice, which were passively transferred with the DENV-immune human sera [22], indicating that the prior DENV humoral immunity enhanced ZIKV infection.

As for the cellular immune response, multiple studies proved that T cell immunity of DENV and ZIKV are cross-reactive [23,24,25,26]. In the *ifnar**1^−/−^* mouse model, depletion of CD8^+^ T cells enhanced ZIKV infection in DENV2-immune mice, and passive transfer of DENV2-immune CD8^+^ T cells protected the naïve mice against ZIKV infection [25]. In the HLA-transgenic mouse model, vaccination with cross-reactive CD4^+^ T cell epitopes of both DENV and ZIKV significantly reduced ZIKV infection, indicating the protective role of CD4^+^ T cells in the cross-reactivity [26].

Prior DENV immunity could protect the fetal mice against ZIKV infection. Fetuses of DENV-immune C57BL/6 mice could maintain normal size when the female mice were challenged with ZIKV during pregnancy, with reduced viral loads in the placenta and fetal mice, while the fetuses of non-immune mice showed atrophy and weight loss [27]. The results were similar in both the *ifnar**1^−/−^* mouse model and the *ifnar**1* transient blockaded mouse model, confirming a protective role of prior DENV immunity to the fetuses [27]. Subsequent CD8^+^ T cell depletion and CD8^+^ T cell passive transfer experiments proved that the cross-reactive CD8^+^ T cells elicited by DENV infection were activated by ZIKV, and aggregated on the placenta, thereby acting as a protective factor. In addition, CD4^+^ T cells also played a supporting role in it [27].

Therefore, cellular immunity might play a more protective role than humoral immunity in sequential infection with DENV and ZIKV.

(3)In vivo non-human primate animal model

In non-human primate animal models, ZIKV infection did not cause obvious changes in their body temperature, body weight, or hematological parameters. Other indicators such as RNA loads and immune responses could be used to decipher the cross-reactivity [4,28,29]. Within a year after primary DENV infection in cynomolgus macaques, the disease severity of the following ZIKV infection was not enhanced or reduced [4]. Consistently, Pantoja, et al. found that pre-existing DENV immunity in rhesus macaques did not result in more severe ZIKV disease, while it induced ADE in vitro [28]. Pre-existing DENV immunity modulated the immune response, presenting a shorter duration of viremia, and fewer activated B cells, CD4^+^ T cells, and CD8^+^ T cells when compared to the naïve control [28].

(4)Study in humans

Cohort studies revealed that neonatal microcephaly caused by ZIKV can occur in people with DENV-immune backgrounds [30]. An ecological study also posed the hypothesis that the elongation of intervals between DENV infection and ZIKV infection reversed the risk of neonatal microcephaly from protection to danger [31]. Despite these ambiguous conclusions, it is more convincing that multitypic DENV infections have a protective effect on newborns against ZIKV infection. Pedroso, et al. collected sera from 29 mothers who gave birth to babies with congenital malformation due to ZIKV infection, and sera of 108 women who gave birth to healthy babies during ZIKV infection served as the control [32]. They corroborated that the seroprevalence of every DENV serotype was lower in mothers with microcephaly babies than that in the control group, and an infection history of two or more DENV serotypes may prevent the development of congenital ZIKV infection syndrome [32]. Consistently, compared with a single infection by DENV, multiple exposures to DENV engendered cross-reactive NAbs responses against ZIKV more robustly [33].

### 3.2. The Impact of Prior ZIKV Immunity on DENV Infection

(1)In vitro serological experiment

Low titers of cross-reactive NAbs to DENV could be detected in sera of C57BL/6 mice and rhesus monkeys with prior exposure to ZIKV, but when the mice were infected twice, no cross-reactive NAbs were detected [4,34]. Incidentally, the same unexpected phenomenon was also seen in the inhibitor ELISA test of DENV cross-reactive antibodies between populations with primary ZIKV infection and populations with sequential infection of DENV–ZIKV [35]. Langerak, et al. collected sera of a population that confirmed ZIKV infection 3 years ago and revealed that ZIKV-immune sera failed to cross-neutralize DENV [36]. Instead, ZIKV-immune mouse sera induced ADE of DENV in THP-1 cells [34]. Stettler, et al. also demonstrated the ADE of DENV with convalescent sera of ZIKV-infected patients [37].

(2)In vivo mouse model

In the *ifnar^−/−^* Sv/129 mouse model, Watanabe, et al. corroborated that when challenged with DENV2, mice primoinfected with ZIKV displayed lower mortality and viral load than the control group [5], suggesting that the immune background of ZIKV could protect mice from DENV infection. While in the *ifnar^−/−^ ifngr^−/−^* mouse model, both passive transfer of murine ZIKV-immune sera and immunization of inactivated ZIKV elevated the mortality when challenged with DENV2 [5], revealing that the humoral immune response elicited by ZIKV was capable of enhancing DENV infection in mice.

## 4. Cross-Reactive Immunity between YFV and DENV

### 4.1. The Impact of Prior YFV Immunity on DENV Infection

(1)In vitro serological experiment

In a study to investigate experimentally how cross-reactive immunity among flavivirus modifies the course of sequential infection, Saron, et al. used YFV-17D to vaccinate immunocompetent mice and found that the immune sera generated high-titer cross-reactive antibodies against DENV1, but failed to cross-neutralize DENV1 [38].

(2)In vivo mouse model

In the mouse model, Saron, et al. confirmed that either the immune history of YFV-17D or passive transfer of YFV-17D immune sera or spleen cells did not affect the DENV1 viral load in the mice’s lymph nodes [38]. Consistently, after the sequential infection of YFV–DENV1, no anamnestic antibody responses to DENV1 were found [38].

However, DENV1 antigens could stimulate the proliferation and activation of splenocytes harvested from the YFV-immune mice, and the numbers of activated CD4^+^ and CD8^+^ effector T cells significantly increased after sequential infection of DENV1 [38], demonstrating that cross-reactive effector T cells induced by YFV-17D immunization were likely to be recalled by DENV1 infection.

(3)Study in humans

In a retrospective study in Brazil, Luppe, et al. evaluated the association between YF vaccination and dengue severity in 11448 lab-confirmed dengue cases. They concluded that YF vaccination did not affect the clinical symptoms of dengue patients [39]. Another team’s work further ascertained that YFV-17D vaccination elicited very limited cross-reactive CD4^+^ and CD8^+^ T cell immunity against other flaviviruses including DENV [40].

### 4.2. The Impact of Prior DENV Immunity on YFV Infection

The finding of Saron’s study displayed that DENV1-immune mouse sera embodied no cross-reactive antibodies or NAbs against YFV, and the proliferation of splenocytes could not be stimulated by YFV-17D antigens [38].

During the outbreak of YF in a jungle garrison and its detachment, Izurieta, et al. collected the sera of 338 survivors (97% of the whole team). They found no relationship between the level of DENV reactive antibodies and the risk for YFV infection but supposed that prior DENV immunity significantly alleviated the severity of the YF disease [41]. However, one limitation is that the DENV reactive antibodies might be due to previous YFV infection rather than DENV infection, so the research cannot accurately reflect the influence of prior DENV immunity on YFV infection.

## 5. Cross-Reactive Immunity between JEV and ZIKV

### 5.1. The Impact of Prior JEV Immunity on ZIKV Infection

(1)In vitro serological experiment

ELISA testing proved that sera of JEV-immune mice or humans contained ZIKV cross-reactive antibodies, and He, et al. demonstrated that the titers of JEV antibodies were positively correlated with the titers of ZIKV cross-reactive antibodies in ZIKV non-endemic areas [6,42]. However, few cross-reactive NAbs against ZIKV were present in JEV-immune sera, and JEV-immune human and mouse sera could induce ADE of ZIKV in vitro [6,43].

(2)In vivo mouse model

By sequential infection of JEV–ZIKV and passive administration of JEV antisera in the *ifnar^−/−^* mouse model, Zhang, et al. reported that the immune history of JE LAV could protect the mice against ZIKV infection without the mediation of cross-reactive antibodies [6]. Passive transfer of highly diluted sera of JE LAV immune humans or rabbits exacerbated ZIKV infection in the *ifnar1* transiently blockaded mouse model [42,43]. Additionally, neonates of JEV-immune female mice showed higher mortality and viremia level than neonates of non-immune mice when challenged with ZIKV [43], suggesting that the anti-JEV IgG antibody from pregnant mice caused ADE of ZIKV in neonates.

Cellular immunity played an important role in the cross-protective effect of prior JEV immunity on the sequential ZIKV infection [44,45]. In the JEV-vaccinated *ifnar^−/−^* adult mouse model, depletion of CD8^+^ T cells increased mortality, weight loss, and severity of clinical symptoms during ZIKV infection, indicating that CD8^+^ T cells were indispensable for the protective effect in the sequential infection of JEV–ZIKV [6]. However, in this research, the passive transfer of JEV immunized CD8^+^ T cells conferred less efficient heterotypic protection against ZIKV infection [6]. In contrast, Chen, et al. demonstrated that the inoculation of JEV-primed CD8^+^ T cells ameliorated the behaviors of one-day-old suckling mice during episodes of ZIKV infection and abrogated the ADE of ZIKV elicited by the antisera of JEV [43]. In a word, JEV-elicited cellular immunity was protective against ZIKV infection, but whether it can offset the ADE effect induced by antibodies is related to the autologous immunological conditions of the mice.

(3)Study in humans

In a phase 1 clinical trial of a ZIKV vaccine, participants preimmunized with the JEV vaccine maintained more durable ZIKV-specific CD4^+^ T cell responses, with more obviousness in conserved epitopes among flaviviruses, suggesting that prior JEV immunity altered the immunogenicity of subsequent ZIKV vaccine [46].

### 5.2. The Impact of Prior ZIKV Immunity on JEV Infection

ZIKV-immune murine sera demonstrated prominent JEV cross-reactive antibody responses. Adoptive transfer of ZIKV-immune sera led to the earlier death of one-day-old suckling mice challenged with JEV than the mock sera, indicating cross-reactive antibodies in anti-ZIKV sera induced ADE of JEV [43]. However, the phenomenon was not found in the pups of ZIKV-immune mice, although the pups maternally acquired cross-reactive antibodies [43]. This may be due to different compositions between antibodies in the sera of ZIKV-immune adult mice and maternally acquired antibodies, which deserves more research.

Passive transfer of ZIKV-immune CD8^+^ T cells displayed a cross-protective effect against JEV infection in the suckling mouse model [43]. This discovery preliminarily revealed the essential role of the cross-reactive cellular immune response in the sequential infection of ZIKV–JEV.

## 6. Cross-Reactive Immunity between JEV and DENV

### 6.1. The Impact of Prior JEV Immunity on DENV Infection

(1)In vitro serological experiment

JEV vaccination elicited high-titer cross-reactive antibodies against DENV1–4 in humans or mice [38,47]. Li, et al. found that both JE LAV and INV immune human sera displayed cross-neutralizing activity against DENV2 and DENV3, and vaccine immune mouse serum also cross-neutralized DENV1–4 slightly [48]. Saron, et al. also detected high-titer DENV1 cross-reactive NAbs in JEV-immune human serum [38].

(2)In vivo mouse model

Prior JEV immunity was cross-protective against DENV in immune-competent mouse models [38,48]. Li, et al. demonstrated that triple vaccination of JE LAV or INV reduced mortality of BALB/c mice challenged with all DENV serotypes [48]. In sequential infection of JEV–DENV1 in C57B/6NTac mice, Saron, et al. detected significantly lower DENV1 viral loads in lymph nodes of the JEV immunized group than that in the non-immune group [38].

Cross-reactive NAbs and cytokines elicited by JE vaccination were boosted during the subsequent DENV challenge, suggesting that prior JEV immunity positively modulated both humoral and cellular immune responses against subsequent DENV infection [38,49]. However, passive transfer of anti-JEV sera significantly elevated DENV1 viral loads in mouse lymph nodes, and passive transfer of JEV immunized T cells had no impact on the episodes of DENV1 infection [38]. The results illustrated that the cross-protective effect in the sequential infection of JEV-DENV1 could not be achieved independently by either the humoral or cellular immune response. The cooperation of both cross-reactive antibodies and T cells would be necessary for the cross-protective effect.

(3)Study in humans

Prior anti-JEV immunity promoted the development of dengue symptoms. In a prospective cohort study conducted in Thailand, Anderson, et al. revealed that among dengue fever patients without anamnestic DENV serology, the positivity of anti-JEV NAbs elevated the probability of symptomatic infection [50]. In addition, among those with multitypic DENV infection histories who were younger than 10 years old, the population with anti-JEV NAbs had an increased proportion of DHF [50].

### 6.2. The Impact of Prior DENV Immunity on JEV Infection

To ascertain the quality of cross-reactive antibodies elicited by various flaviviruses, Saron, et al. tested IgG and NAbs levels of sera of flavivirus-immune immunocompetent mice, and proved that DENV-immune sera were slightly cross-reactive against JEV while with no cross-neutralizing ability [38].

In the human study, an analysis of clinical data of 182 JE cases revealed that patients who were dengue IgG-positive performed better in clinical outcomes and prognoses than dengue IgG-negative patients [51], suggesting that prior DENV immunity was cross-protective against JEV infection.

## 7. Cross-Reactive Immunity between JEV and WNV

### 7.1. The Impact of Prior JEV Immunity on WNV Infection

(1)In vitro experiment

JE-vaccinated human sera exhibited cross-reactivity against WNV. Yamshchikov, et al. immunized seven volunteers with JE INV and YFV-17D, monitoring their seroconversion for four years [52]. They demonstrated that the sera had high titers of IgG and low titers of NAbs that were cross-reactive against WNV. All anti-WNV responses followed the changing trend of anti-JEV responses, especially after a boost with JE vaccines [52], revealing that the two vaccines induced cross-reactive immunity to WNV, and JE INV occupied a great territory in it.

However, in the mouse model, other research demonstrated that JE INV induced low titers of WNV cross-reactive NAbs only in the presence of adjuvants [49]. Similarly, Tang, et al. did not find adequate WNV cross-reactive NAbs in the sera collected from JEV immunized or infected people [53]. These results revealed the limited ability of prior JEV immunity to induce WNV cross-reactive NAbs.

(2)In vivo mouse model

Experiments in various mouse models revealed that prior JEV immunity promoted cross-protection against WNV infection [54,55]. In a hamster model, former JEV infection significantly reduced mortality and viremia level of mice challenged with WNV [56]. In BALB/c and C3H/HeN mouse models, similar cross-protective effects were demonstrated in the vaccination of JE INV or JEV envelop domain III (EDIII) protein [57,58]. In a C57BL/6 mouse model, JE INV accompanied by Advax-adjuvant could harness the maximum potential in the cross-protection against WNV infection [59].

Cellular immunity was less important in the cross-protection. In sequential infection of JEV–WNV, Petrovsky, et al. observed identical cross-protection between β2-microglobulin-deficient mice and immune-competent mice, confirming the needlessness of CD8^+^ T cells in the cross-protective effect [59]. However, passive transfer of JEV immune B cells or CD4^+^ T cells to C57BL/6 mice significantly ameliorated their mortality and weight change during WNV infection [59], demonstrating that immune cells associated with humoral immunity positively modulated the cross-protection against WNV infection.

### 7.2. The Impact of Prior WNV Immunity on JEV Infection

The considerable sequence identity created antigenic similarities between JEV and WNV [60]. Nowadays, studies have further authenticated that regimens of WNV vaccination conferred cross-protection against JEV infection. Martina, et al. demonstrated that double inoculation of inactivated WNV or WNV EDIII protein in C57BL/6 mice elicited low-titer JEV cross-reactive NAbs, and significantly ameliorated death and weight loss during JEV infection [61]. Additionally, vaccination with WNV NS1 protein also generated cross-protection against JEV infection [62].

## 8. Discussion

*Flaviviridae* viruses spread worldwide and have a far-reaching effect on human life. Investigating the interactions among flaviviruses can provide theoretical clues for the prediction of flavivirus epidemics and the development of effective vaccines. We retrospected the research of cross-reactivity among several important mosquito-borne flaviviruses in detail. Prior flavivirus immunity mentioned above achieved cross-reactive effects on resisting subsequent infection of heterologous flaviviruses besides the YFV (Table 2), which is consistent with the low sequence identity between YFV and the other flaviviruses (Table 1).

At present, there are still some limitations in the test of cross-reaction, especially in the evaluation of ADE. Although Fcγ receptor-bearing cells have been widely used to measure the enhancement of flavivirus infection [34,37], reactions in vitro do not always reflect what is happening in vivo [5]. To make their comparison more intuitive, we summarized the in vivo cross-reactive immunities among the five flaviviruses in Table 2. Preexisting immunities play distinct roles in subsequent viral pathogenesis and even outcome. The studies conducted on different models always draw different conclusions: prior JEV immunity offered cross-protection against DENV in mice [38], while a contrary conclusion was obtained in the population study [50] but, as an acknowledged limitation, there is no ideal mouse model to investigate the ADE [63]. Therefore, no symptom or complete survival in the mouse model does not signify that sequential infections of flaviviruses are dependable for human beings [5]. In addition, the results of inbred mice with a highly homologous genetic background cannot entirely reflect the complex situation in humans.

The cross-reaction depends on the balance between cross-reactive humoral immunity and cross-reactive cellular immunity, and the equilibrium point in the tug of war varies with the immunological conditions of the biosome. However, immune-competent mouse or non-human primate models are not susceptible to some flaviviruses, such as DENV and ZIKV. The relevant studies [5,6,25,27,43,45] mainly utilize immune-compromised or suckling mice, which could not correctly reflect the immune-competent situation. Therefore, to further advance the research progress of cross-reaction among flaviviruses, it is necessary to develop a more suitable animal model.

In recent years, demographic characteristics demonstrated that fewer severe dengue cases in China were observed than that in other dengue hyperendemic countries [64]. Meanwhile, severe dengue cases occurred in adults rather than children, which is also different from other countries in Southeast Asia [64,65]. It is presumed that JE vaccination in mainland China for decades contributed to the cross-reactive protection against DENVs and alleviated disease severity, and this effect in mice has been confirmed in our studies [48,66]. In addition, our team has also been devoted to elucidating the cross-reactivity of JE vaccination on the ZIKV disease. In the mouse model, we have proven the cross-protective role of JE LAV against ZIKV infection and believe that cellular immunity primed by JE vaccination predominantly mediated the cross-protection [45].

Co-circulation of different flaviviruses has made people living in the epidemic regions suffer a lot. It is of great significance to figure out the impact of preexisting cross-reactive immune responses upon subsequent flavivirus infections. These studies provoke cautions for the potential risk of enhancement in sequential infections, especially in ZIKV possible epidemic regions. Additionally, the development of diagnostic reagents for flavivirus infections should avoid cross-reactive epitopes to improve the specificity. For the invention of vaccines, sub-neutralizing antibodies that bind to poorly accessible epitopes need to be carefully eschewed [67,68]. It will be of great practical significance and scientific value to further study the mechanism of cross-reaction among flaviviruses, especially the function of the cellular immune response. These efforts will be informative for the design and adoption of effective vaccines against flaviviruses to prevent cross-enhancement and target cross-protection.

## Figures and Tables

**Figure 1 viruses-14-01213-f001:**
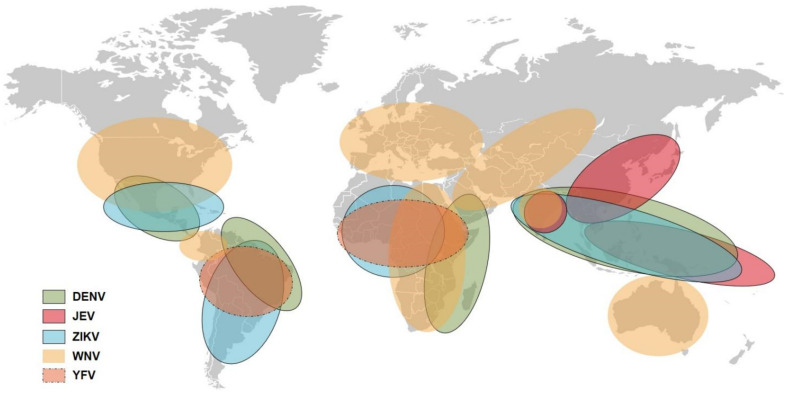
Global distribution of five mosquito-borne flaviviruses. The figure was created based on the epidemiological data of the WHO and the American CDC (Centers for Disease Control and Prevention).

**Table 1 viruses-14-01213-t001:** Identity analysis of amino acid sequences based on the polyprotein and the envelop protein of five mosquito-borne flaviviruses.

	WNV (%)		ZIKV (%)		YFV (%)		DENV1 (%)		DENV2 (%)		DENV3 (%)		DENV4 (%)	
	PP	E	PP	E	PP	E	PP	E	PP	E	PP	E	PP	E
JEV	77	79	56	56	45	44	51	51	51	48	51	49	51	48
WNV			57	56	45	44	51	52	52	49	51	48	52	50
ZIKV					46	43	55	58	55	54	56	58	56	56
YFV							45	43	45	44	45	42	45	40
DENV1									72	69	78	78	69	64
DENV2											72	69	70	64
DENV3													70	63

Note: 1. PP: polyprotein; E: envelop protein. 2. The multiple sequence alignment in this table is created by the AlignX sequence alignment program in vector NTI advance 11.5. 3. The serial numbers in GenBank are as follows: Polyprotein: NP_059434.1 (JEV, strain JaOArS982), NP_041724.2 (WNV, strain 956), YP_002790881.1 (ZIKV, strain MR766), AHB63685.1 (YFV, strain Asibi), NP_059433.1 (DENV1, strain Nauru Island, Western Pacific), NP_056776.2 (DENV2, strain 16681), YP_001621843.1 (DENV3, strain D3/H/IMTSSA-SRI/2000/1266), NP_073286.1 (DENV4). Envelop protein: NP_775666.1 (JEV, strain JaOArS982), NP_776014.1 (WNV, strain 956), YP_009227198.1 (ZIKV, strain MR766), NP_740305.1 (YFV, 17D vaccine strain), NP_722460.2 (DENV1, strain Nauru Island, Western Pacific), NP_739583.2 (DENV2, strain 16681), YP_001531168.2 (DENV3, strain D3/H/IMTSSA-SRI/2000/1266), NP_740317.1 (DENV4).

**Table 2 viruses-14-01213-t002:** The in vivo effect and probable mechanism of cross-reactive immunity on sequential infection among five mosquito-borne flaviviruses.

	Model	Effect	Probable Mechanism
DENV–ZIKV	mouse	protection [5]	cross-reactive sera: no effect/enhancement [22] cross-reactive T cells: protection [25,26]
	human	multiple exposures: protective effect for newborns [32]	high titers of cross-reactive NAbs [33]
ZIKV–DENV	mouse	previous infection: protection [5] INV vaccination: enhancement [5]	cross-reactive sera: enhancement [5] cross-reactive T cells: deserve more research
YFV–DENV	mouse	not observed [38]	cross-reactive sera: no cross-neutralizing ability [38] cross-reactive T cells: fail to protect [38]
	human	no effect on the clinical symptoms [39]	cross-reactive CD4^+^ and CD8^+^ T cells: limited [40]
JEV–ZIKV	mouse	protection [6,45]	cross-reactive sera: enhancement [42,43] cross-reactive T cells: protective, but depended on the immune conditions [6,43,45]
ZIKV–JEV	mouse	depend on more investigations	cross-reactive sera: enhancement [43] cross-reactive T cells: protection [43]
JEV–DENV	mouse	protection [38,48]	cooperation of cross-reactive antibodies and T cells [38]
	human	increase the probability of symptomatic infection [50]	JEV antibody is associated with occurence of symptom [50].
DENV–JEV	human	dengue IgG-positive patients have a significantly better outcome [51]	not reported
JEV–WNV	mouse	previous infection: protection [56] INV or protein vaccination: protection [57,58,59]	cross-reactive CD8^+^ T cells: little effect [59] cross-reactive B or CD4^+^ T cells: protection [59]
WNV–JEV	mouse	INV or protein vaccination: protection [61,62]	not reported

## Data Availability

All data generated and analyzed in this research are included in the article.

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
