# Peer review of "Cross-Reactive Immunity among Five Medically Important Mosquito-Borne Flaviviruses Related to Human Diseases"

_viruses, 2022, doi:10.3390/v14061213_

Round 1
Reviewer 1 Report
In this review, the authors discussed the cross-reactive immunity among five medically important mosquito borne flaviviruses (JEV, DENV, ZIKV, WNV and YFV). Overall, the information provided by the manuscript is valuable to the field. The manuscript is well organized and clearly written.
I have the following suggestion for the authors: added the homology analysis of E gene sequence in Table 1.
Reviewer 2 Report
This review reports on the state of the art on 5 flaviviruses of medical interest and on the consequences of cross-reactions induced by these different viruses during coinfections, either protective effects or, on the contrary, promoting the infection by another flavivirus. This review deals with a very interesting subject and a map of the world in which these 5 flaviviruses are present would have provided a better understanding of the risk of co-infection by these viruses, or at least a reference in which such a figure is presented.
Major comments:
In general, this review does not stand on its own and it is necessary to look for explanations in the articles cited.
Summarizing the in vivo cross-reactive immunities among the 5 flaviviruses in a table to make their comparison mor intuitive was an excellent idea but this table should be self-explanatory. Indeed, the row corresponding to JEV-DENV in the mouse model and mentioned as protection in the history of vaccination or infection, enhancement of humoral immunity and no impact on cellular immunity is difficult to understand just by reading the table.
Moreover, the concordance between Table 2 and the text is not always clear. As an example, lines 178-185 and table 2: The authors present a two-stage study on mice: in the first stage, mice primoinfected with ZIKV (is that what ZIKV-immune mice means?) showed less signs of morbidity and mortality when they were subsequently infected with DENV. In a second step, mice are inoculated with murine ZIKV-immune sera and inactivated ZIKV and then infected with DENV, an exacerbation of the DENV infection is observed. For this study reported in Table 2, the authors indicate that a protection in vaccination of infection history and an exacerbation in humoral immunity, the link is difficult to make without reading the text.
Minor comments:
line 34 : ixodids are ticks, it is not necessary to mention both
lines 36-37: “and WNV, and ZIKV”: The first “and” should be removed
line 38: “ZIKV has transmitted” : please could you correct by has been transmitted
Line 39: the authors could also have mentioned the expansion of WNV in the United States following its introduction in New York in 1999 and the subsequent epidemic.
Lines 57 to 60: “Additionally, other vector-borne flaviviruses including tick-borne encephalitis virus and Usutu viru, … have been reported to cause cross-protective immunity”. This sentence is quite difficult to understand. Could the authors clarify whether they mean that TBEV infection can cause protective cross-immunity in an individual who is infected with USUV? Or whether TBEV or USUV infection can cause protective cross-immunity in an individual who is infected with DENV, ZIKV, WNV, JEV or YFV?
Line 89 : “region of EDII”: the abbreviation EDII should be explained
Lines 136-137 : From the studies they present, I understand that cellular immunity would have a more protective role than humoral immunity but not that cellular immunity has a more important role.
Line 290: “homural” is misspelled.
Table 2: Is reference 21 of the DENV-ZIKV human model mentioning the protection of newborns correct? would not be the reference 31 ?
